# Genomic Surveillance of Rabies Virus in Georgian Canines

**DOI:** 10.3390/v15091797

**Published:** 2023-08-24

**Authors:** Celeste Huaman, Adrian C. Paskey, Caitlyn Clouse, Austin Feasley, Madeline Rader, Gregory K. Rice, Andrea E. Luquette, Maren C. Fitzpatrick, Hannah M. Drumm, Lianying Yan, Regina Z. Cer, Marina Donduashvili, Tamar Buchukuri, Anna Nanava, Christine E. Hulseberg, Michael A. Washington, Eric D. Laing, Francisco Malagon, Christopher C. Broder, Kimberly A. Bishop-Lilly, Brian C. Schaefer

**Affiliations:** 1Department of Microbiology, Uniformed Services University, Bethesda, MD 20814, USA; 2Henry M. Jackson Foundation for the Advancement of Military Medicine, Bethesda, MD 20814, USA; 3Genomics and Bioinformatics Department, Biological Defense Research Directorate, Naval Medical Research Command-Frederick, Fort Detrick, Frederick, MD 21702, USA; 4Leidos, Reston, VA 20190, USA; 5State Laboratory of Agriculture (SLA), Tbilisi 0159, Georgia; 6US Army Medical Research Directorate-Georgia (USAMRD-G), Tbilisi 0198, Georgia; 7Dwight D. Eisenhower Army Medical Center, Augusta, GA 30905, USA

**Keywords:** rabies, lyssaviruses, RABV, dog, jackal, canine, neutralization, genomics, next-generation sequencing, phylogeny

## Abstract

Rabies is a fatal zoonosis that is considered a re-emerging infectious disease. Although rabies remains endemic in canines throughout much of the world, vaccination programs have essentially eliminated dog rabies in the Americas and much of Europe. However, despite the goal of eliminating dog rabies in the European Union by 2020, sporadic cases of dog rabies still occur in Eastern Europe, including Georgia. To assess the genetic diversity of the strains recently circulating in Georgia, we sequenced seventy-eight RABV-positive samples from the brain tissues of rabid dogs and jackals using Illumina short-read sequencing of total RNA shotgun libraries. Seventy-seven RABV genomes were successfully assembled and annotated, with seventy-four of them reaching the coding-complete status. Phylogenetic analyses of the nucleoprotein (N) and attachment glycoprotein (G) genes placed all the assembled genomes into the Cosmopolitan clade, consistent with the Georgian origin of the samples. An amino acid alignment of the G glycoprotein ectodomain identified twelve different sequences for this domain among the samples. Only one of the ectodomain groups contained a residue change in an antigenic site, an R264H change in the G5 antigenic site. Three isolates were cultured, and these were found to be efficiently neutralized by the human monoclonal antibody A6. Overall, our data show that recently circulating RABV isolates from Georgian canines are predominantly closely related phylogroup I viruses of the Cosmopolitan clade. Current human rabies vaccines should offer protection against infection by Georgian canine RABVs. The genomes have been deposited in GenBank (accessions: OQ603609-OQ603685).

## 1. Introduction

Rabies virus (RABV), the prototypical lyssavirus, and rabies-related viruses are neurotropic pathogens that cause rabies, a uniformly fatal encephalitic disease of both humans and a diversity of mammals. Rabies remains a neglected tropical disease and is estimated to cause approximately 60,000 human fatalities per year, the majority occurring in children across Africa and Asia [1]. The annual global economic costs of animal and human deaths caused by rabies has been estimated to be in the range of tens of billions of U.S. dollars [2]. Although wealthy countries have largely eliminated rabies via a combination of effective animal vaccination programs and wide access to post-exposure prophylaxis (PEP) for known or potential exposures to rabid animals, such preventive measures are not widely available in much of Africa and Asia. Additionally, there are no effective treatments for symptomatic rabies [1]. Notably, domestic dogs are responsible for 99% of transmission events to humans and subsequent rabies fatalities [3].

Highly successful animal vaccination programs in resource-rich countries have largely eliminated dog-mediated rabies in the Americas and Western Europe (although persistence occurs in wildlife sources, including bats, raccoons, and foxes). However, bat-mediated rabies is becoming a larger source of rabies disease in the Americas [4,5] since the effective interruption of dog-mediated transmission. The goal of *Zero by 30*, a program promoted by the United Nations, the World Health Organization, and several other international health organizations, is to eliminate dog-mediated rabies deaths by 2030 [6]. Although a goal of the European Union (EU) was to eliminate dog rabies by 2020 [7], this goal has not been met, as evidenced by the fact that sporadic cases of dog-mediated rabies still occur in Eastern Europe [8]. Indeed, Georgia and other Eastern European countries are in close proximity to the Middle East and West Asia, and rabies remains endemic in most countries in this region. Within rabies-endemic countries, rabid dogs transmit rabies to local wildlife, such as jackals. Rabid jackals may then cross borders and transmit rabies to domestic dogs in Georgia and other Eastern European countries in which rabies elimination programs are ongoing. Additionally, local efforts to eliminate rabies in Eastern European countries may not adequately cover rural, sparsely populated regions that may be subjected to a continuous reintroduction of rabies from neighboring countries [9,10]. Thus, the complete elimination of rabid canines from Eastern Europe is a difficult goal to achieve without a simultaneous concerted effort to eliminate dog rabies from rural regions of these countries as well as in bordering countries in the Middle East and Western Asia [11]. Given the ongoing cycles of rabies transmission from wildlife to domestic dogs and other animals in Eastern European countries, detailed information regarding the genetic diversity of rabies strains currently circulating in canines in this geographic region is of high relevance.

Lyssaviruses are negative-stranded RNA viruses of the family *Rhabdoviridae*. The majority of lyssavirus species are classified as belonging to one of two phylogroups (phylogroups I and II), with a few divergent species remaining ungrouped. RABV and most other lyssavirus species reported to infect humans are members of phylogroup I. Various species of canines are the primary animal hosts of RABV, whereas, specific species of bats are the primary hosts for the other lyssaviruses [12]. The transmission of RABV to humans occurs following a bite or scratch from a rabid animal. RABV generally replicates in muscle cells before transiting into peripheral nerves at the neuromuscular junction. RABV then uses these afferent nerves to enter the spinal cord, where another round of viral replication occurs, followed by ascension to the brain; the death of the host occurs via encephalitis. Following further replication in the brain, RABV spreads to multiple peripheral tissues via efferent nerves. The spread to and replication within salivary glands enables transmission to subsequent hosts [12,13].

RABV genomes encode five genes, *N* (nucleoprotein), *P* (phosphoprotein), *M* (matrix), *G* (glycoprotein), and *L* (polymerase). Among these genes, *N* and *G* have been most characterized at the level of nucleic acid sequence [14]. Because RABV, like other RNA viruses, is replicated by an error-prone RNA-directed RNA polymerase, mutations accumulate over relatively short periods of time [15]. However, there is also selective pressure against many non-synonymous mutations, and the evolution of RABV over time is thus slow [16]. Canine RABV is dispersed across most of the world, with genomic sequences segregated into six major phylogenetic clades. The Cosmopolitan clade is globally distributed, whereas other clades are more regional. The Africa-2 clade is found in west Africa; the Africa-3 clade predominates in east Africa; the Asian clade is primarily in central and eastern Asia; the Arctic cluster is found mostly in regions bordering the arctic circle and in western Asia; and the Indian Subcontinent clade is present in Sri Lanka and southern India [16,17,18].

To date, the majority of the phylogenetic studies of RABV isolates from infected hosts have focused either on limited sequencing of a large number of isolates (e.g., *N* only) or on full genome sequencing of small numbers of isolates. Although a few studies have provided complete sequences of a large number of isolates [16,17,19,20], we are aware of no such studies that have reported complete sequences of large numbers of samples collected in Eastern Europe. With regard to Georgia and its surrounding countries, the great majority of publicly available RABV sequences exist primarily as population sets of the *N* gene deposited to NCBI [21]. In this work, we not only expanded existing knowledge of the diversity of the *N* gene but also evaluated the diversity among publicly available complete RABV genome sequences. To conduct this study, brain tissues from rabid dogs and jackals from Georgia, Caucasus region, were collected during the years 2018–2021, and RABV PCR-positive samples were processed for genome sequencing. RNA was extracted from 78 selected tissue samples for sequencing and genomic analysis. These analyses resulted in the production of 74 complete plus 3 quasi-complete RABV sequences from Georgian canines. Herein, we provide complete RABV sequences recovered from approximately 80 canines in Georgia, including eight sequences from jackals. We relate the distribution of identified amino acid variations to known antigenic sites in the G ectodomain, and we test the ability of a potent human anti-G monoclonal antibody (mAb) to neutralize three cultured isolates.

## 2. Materials and Methods

### 2.1. Biosafety Precautions

As RABV is a lethal virus that can be transmitted by contact (e.g., through mucous membranes or damaged skin) or via inhalation, BSL-2 containment practices were followed for all steps in which virus-infected tissues, viral stocks, or virally infected tissue culture cells were handled. All laboratory manipulations with RABV were performed in a biosafety cabinet by rabies-vaccinated personnel using appropriate BSL2 procedures for lyssaviruses as outlined in the *CDC Biosafety in Microbiological and Biomedical Laboratories (BMBL) 6th Edition* [22], following protocols approved by the USU Institutional Biosafety Committee.

### 2.2. Collection of Brain Tissue from Suspected Rabid Dogs and Jackals in Georgia

As part of an ongoing rabies surveillance program conducted by the State Laboratory of Agriculture (SLA) in Tbilisi, Georgia, brain tissue was collected from dogs and jackals suspected of being rabid, based on observation of disease signs consistent with rabies. Brain tissue was harvested from these canines as soon as possible post-mortem. Brain tissue was divided into portions of approximately 1 g, which were placed into labeled cryovials. Brain tissue was then frozen and stored on dry ice or in a −80 °C freezer until the time of RNA extraction or viral culture. Each sample was given an identifier, RABies Virus-GEOrgia-x (RABV-GEO-x), in which x is a unique number between 1 and 100. Tissues were shipped on dry ice to Uniformed Services University for further analysis, under the following importation permits 20210830-3180A, issued by the U.S. Centers for Disease Control and Prevention (CDC); and 143204, issued by the U.S. Department of Agriculture.

### 2.3. RNA Purification and Degenerate PCR Amplification and Sequencing of the Lyssavirus N Gene

Total RNA was prepared by placing approximately 100 mg of frozen brain tissue into a 5 mL Eppendorf Safe-Lock tube with TRIzol (product number 15596026, Invitrogen, Waltham, MA, USA) and zirconium oxide lysis beads (Next Advance, ZROB20-RNA). The brain tissue was homogenized using a BulletBlender (model BB5E-AU, Next Advance, Troy, NY, USA) within a biosafety cabinet. The homogenate was then transferred to 1.5 mL microcentrifuge tubes to produce total RNA, following the TRIzol RNA purification protocol, as per the manufacturer (Invitrogen, Waltham, MA, USA). To degrade any contaminating DNA, RNA samples were treated with RQ1 DNAse (Promega, Madison, WI, USA), following the manufacturer’s protocol. After phenol extraction and ethanol precipitation, total RNA was resuspended and quantified using a NanoDrop One spectrophotometer (Thermo Scientific, Waltham, MA, USA).

Samples were first screened to identify RABV-positive samples and to determine whether sequences were consistent with RABV strains of Georgian origin [21]. For this purpose, we used a modified degenerate nested reverse-transcriptase PCR protocol to amplify a region of the *N* gene, a region that codes for approximately the N-terminal half of the protein. This protocol was chosen because it can successfully amplify a wide array of lyssavirus *N* genes, including those outside of phylogroup I [23]. The JW12 primer (ATGTAACACCYCTACAATTG) was used for first-strand cDNA synthesis, followed by primary PCR, using JW12 and an equimolar mix of primers JW6(DPL), JW6(E), and JW6(M) (CAATTCGCACACATTTTGTG; CAGTTGGCACACATCTTGTG; and CAGTTAGCGCACATCTTATG, respectively). The secondary PCR was performed with primers modified by the addition of binding sites for sequencing primers M13R49 and M13F43. In this manner, secondary PCR was performed with primer M13R49-JW12 (GCTGAGCGGATAACAATTTCACACAGGATGTAACACCYCTACAATTG), and an equimolar mix of M13F43-JW10(DLE2), M13F43-JW10(ME1), and M13F43-JW10(P) (GCTAGGGTTTTCCCAGTCACGACGTTGTCATCAAAGTGTGRTGCTC; GCTAGGGTTTTCCCAGTCACGACGTTGTCATCAATGTGTGRTGTTC; and GCTAGGGTTTTCCCAGTCACGACGTTGTCATTAGAGTATGGTGTTC, respectively). Samples yielding visible PCR products of the predicted size (~635 bp) were sequenced from both ends using primers M13R49 (GAGCGGATAACAATTTCACACAGG) and M13F43 (AGGGTTTTCCCAGTCACGACGTT) via Sanger sequencing (Eurofins Genomics, Louisville, KY, USA). Primer sequences were removed, and the remaining sequence of each isolate (approximately 542 bp) was subjected to phylogenetic relatedness analysis using Geneious Prime software (Dotmatics, Boston, MA, USA). Briefly, after removing primer sequences from each read, forward and reverse sequences were aligned to create a consensus sequence for each clone. The RABV-GEO *N* gene PCR product sequences were then compared to *N* gene sequences from a variety of known GenBank clones (representing available NCBI *N* sequences from Georgia and other geographically proximal countries) using the Geneious Tree Builder function, employing the Tamura-Nei genetic distance model, the neighbor-joining tree build method, with no outgroup.

### 2.4. Shotgun Sequencing of Total RNA Samples

Samples determined to contain RABV genomes based on the above sequencing of PCR products were next analyzed via next-generation sequencing (NGS) methods in order to determine the full genomic sequence of each RABV isolate. Total RNA was used to prepare shotgun libraries for sequencing using the NEBNext Ultra II RNA Library Prep for Illumina (New England Biolabs; Ipswich, MA, USA) following the manufacturer’s instructions. Briefly, 5 μL of RNA at ~50 ng/μL was first fragmented and reverse transcribed into ssDNA. The ssDNA was then used as template for the synthesis of the complementary DNA strand to obtain dsDNA. The dsDNA was end-repaired and 3′-end extended to add dA overhangs. Hairpin sequencing adaptors, containing 5′-dT overhangs and a U ribonucleotide in the hairpin loop, were added by DNA ligation and subsequently cleaved at the U sites. The libraries were then amplified and indexed by PCR using NEBNext Unique Dual Indexes. Prior to sequencing, the libraries were evaluated for quality using Agilent D1000 kit (Agilent Technologies; Santa Clara, CA, USA). The libraries that passed QC were then quantified using Qubit dsDNA BR assay (ThermoFisher Scientific; Waltham, MA, USA) and pooled for sequencing. An initial set of 4 samples (RABV-GEO-6, -9, -13 and -20) were sequenced using a MiSeq Reagent Kit v3 600 cycle and a MiSeq sequencer (Illumina; San Diego, CA, USA). The remaining libraries were sequenced using a NovaSeq6000 S4 Reagent Kit v1.5 300 cycles sequencing kit and a NovaSeq6000 sequencer (Illumina; San Diego, CA, USA).

### 2.5. Sequencing-Data Processing and Genome Assembly

Raw reads were trimmed and filtered prior to assembly. Briefly, bbtools v39.01 suite [24] was used to trim raw reads using bbduk and to map quality-controlled reads >Q20 to Lyssavirus taxonid 11286, downloaded from NCBI using taxonkit [25]. Positively filtered lyssavirus reads were subsequently assembled using metaSPAdes v3.15.3 [26]. Assembly quality and query cover were evaluated using Bandage v0.8.1 [27]. Contigs that had sparse coverage of the genome were abandoned. For samples that did not produce complete RABV genomes using metaSPAdes, reads were further subsampled to 3000–25,000 paired-end reads using bbtools v39.01 and assembled using Unicycler v0.5.0 [28]. When appropriate, manual genome closure was performed with evidence supported by contigs assembled using both metaSPAdes and Unicycler. All genomes were manually reviewed for quality and annotated using an ORF finder in CLC Genomics Workbench v23 (QIAGEN).

### 2.6. Analysis of Heterozygous Single Nucleotide Variants (SNVs)

To evaluate the possibility of coinfection with multiple genotypes or lineages of rabies virus, trimmed reads were mapped back to assemblies requiring at least half the read to map with a minimum of 80% identity and evaluated for heterozygous SNVs using CLC Genomics Workbench v23 (QIAGEN, Hilden, Germany) with a minimum frequency of 35% and a minimum coverage of 10.

### 2.7. Phylogenetic Analysis

Alignments of the open reading frames predicted for *N* and *G* genes each were generated using CLC Genomics Workbench v23 (QIAGEN, Hilden, Germany). To generate maximum likelihood trees, IQ-TREE v2.0.3 [29,30] was used with 1000 bootstraps [31] and visualized using FigTree v1.4.4 [32]. In both cases, the best-fit model was TVMe + G4.

### 2.8. Sequence Analysis of the G Glycoprotein Ectodomain

Amino acid sequences of the *G* genes were obtained by direct translation of the genomic assemblies using ORFfinder [33]. Predicted G glycoprotein sequences were aligned with Clustal Omega [34] and trimmed to eliminate the N-terminal signal peptide, the C-proximal transmembrane domain, and the C-terminal cytoplasmic domain [35,36]. The resulting 439 aa ectodomains were aligned using Clustal Omega and the resulting alignments and phylograms were used to group samples with identical G glycoprotein ectodomain sequences and to obtain the consensus sequence.

### 2.9. Principal Components Analysis (PCA) of Ectodomain Group and Geographic Origin of Samples

R function ‘prcomp’ was used to perform a principal components analysis on a data matrix representing ectodomain group assignments and the subregion of Georgia from which each sample was collected [37]. Results were visualized using ggplot2 [38].

### 2.10. Clustering of RABV Genomes and RABV Protein Sequences

Full length genomes of RABV, as well as amino acid sequences for each gene product, were downloaded from NCBI (taxID: 11292, Lyssavirus rabies; accessed March 2023) [39] and combined with the RABV-GEO sequences. Amino acid sequences were subsequently filtered to include only full or nearly full-length gene products. Genome and polypeptide sequences were then clustered using MMseqs2 v13.45111 [40], specifying a minimum of 0.95 and 0.8 sequence identity, respectively. A target clustering mode was utilized so that partial sequences were heavily weighted against being representative of a cluster.

### 2.11. Alignment of Clustered RABV Amino Acid Sequences

Representative amino acid sequences were extracted for each cluster and aligned using CLC Workbench v23 (QIAGEN; Hilden, Germany), using default parameters (gap open cost = 10; gap extension cost = 1). RABV-GEO-97, selected automatically by MMseqs2 as the representative for P cluster 8, was also included as an additional representative in N, M, G, and L alignments. BlastP [39] was used to identify conserved domains from the Pfam database [41], and the relevant positions were subsequently evaluated for the degree of conservation among representative sequences.

### 2.12. Analysis of Predicted Epitopes for RABV-GEO P, M, L Proteins

RABV epitopes within L, M, and P amino acid sequences were identified using resources from the IEDB-AR (immune epitope database) [42], accessible through the Bacterial and Viral Bioinformatics Resource Center (BV-BRC) [43].

### 2.13. RABV Isolation and Neutralization with mAb A6

To isolate RABV, frozen brain tissue was homogenized into a 20% [*w*/*v*] suspension in 1 mL of sterile 1× PBS using a Bullet Blender (Next Advance, model BB5E-AU). This homogenate was then clarified by centrifugation. Next, 0.5 mL of clarified supernatant was transferred to a 50 mL disposable centrifuge tube containing 10^6^ N2a cells (a mouse neuroblast cell line, ATCC CCL-131), followed by 30 min incubation in a 37 °C, 5% CO_2_, with occasional agitation. Complete DMEM (10 mL) was then added, and tubes were centrifuged for 15 min at 700× *g*. The growth media was replaced with 15 mL of fresh DMEM; cells were resuspended and aliquoted to one well of a 96-well plate (75 μL), with the remainder transferred to a T75 flask. After 48 h, the cells in the 96-well plate were stained with FITC anti-Rabies G (Cat #800-092, Fujirebio, Malvern, PA, USA) at a dilution of 1:100 overnight at 4 °C to identify the foci of RABV-infected cells. At approximately 72 h post-infection, low-titer viral supernatant and half of the cells from the T75 flask were cultured with fresh N2a cells (5 × 10^5^), again distributing between one well of a 96-well plate and a T75 flask. Viral supernatant from this second passage was concentrated by ultracentrifugation, followed by titering on N2a cells.

Neutralization with anti-lyssavirus G human monoclonal antibody A6 [44] was performed with the titered RABV-GEO viruses. Viral stocks at an MOI of 0.1 were incubated with 2-fold serial dilutions of A6 at 37 °C in a 5% CO_2_ incubator for 90 min. The virus/A6 mix was transferred to the corresponding wells with N2a cells (4 × 10^4^ cells/well) in a 96-well plate. Plates were incubated at 37 °C in a 5% CO_2_ incubator for 48 h. Cells were fixed with 4% paraformaldehyde for 1 hr, followed by washing (1× PBS) and permeabilization (1× PBS, 0.2% Triton X-100, 0.1% sodium azide) for 10 min at ambient temperature. Following further washing, wells were blocked for 20 min at ambient temperature (DMEM with 10% fetal bovine serum). Staining was then performed with FITC anti-Rabies G (Cat #800-092, Fujirebio, Malvern, PA) at a dilution of 1:100 overnight at 4 °C. Following washing, foci were counted using an ELISpot analyzer (model S6 Flex M2, Immunospot, Shaker heights, OH).

## 3. Results

### 3.1. PCR Screening and Sequencing of N genes of Georgian Canine Brain Tissue Isolates

To gain further insight into the population of rabies viruses recently circulating in Eastern Europe, we analyzed a large group of canine samples collected in Georgia. Primary brain tissue isolates from suspected rabid dogs and jackals were processed to yield total RNA. Purified RNAs were then screened using nested degenerate PCR primers that amplify approximately the N-terminal half of the lyssavirus *N* gene [23]. The majority of isolates yielded a clear PCR product of the predicted molecular weight. Using the M13R43 and M13F49 sequencing primer binding sites that were incorporated into the *N* gene amplification primers, the PCR products were sequenced to provide an initial assessment of the viral genotype. Phylogenetic analyses of these partial *N* gene sequences suggested that the viruses were closely related to Eastern European isolates, consistent with their Georgian origin (Appendix A). To better clarify the genetic relatedness to other strains, we proceeded to fully sequence these isolates.

### 3.2. Sequencing and Assembly of Seventy-Seven Complete Georgian RABV Genomes

The majority of RABV sequences in publicly available databases such as NCBI are partial genome sequences, generally covering a portion or all of the *N* gene or *G* gene. To provide more complete information regarding the sequence diversity of RABVs circulating in Georgia, we sequenced, assembled, and annotated 77 total RNA samples (Appendix A). Based on the proportion of viral reads in the total RNA, the average load of RABV RNA in the brain tissue samples was approximately 0.2%. The depth of coverage ranged from 50 to 2000-fold for most of the samples. RABV-GEO-100 was positive for the presence of the RABV virus judging by the sequencing results but had a low depth and sparse coverage of the genome and therefore was not used in further analyses. RABV genomes were successfully assembled using metaSPAdes for seven samples (RABV-GEO-6, 9, 13, 20, 74, 78, and 97), while the assembly of the rest of the samples with Unicycler resulted in an additional set of 64 successfully assembled samples. An additional set of six samples (RABV-GEO-23, 30, 36, 54, 57, and 62) produced complete genomes when manually closed. From a total of 77 successfully assembled genomes, 74 samples were fully annotated for all five rabies virus genes (*N*, *P*, *M*, *G*, and *L*), making them coding complete, while for the other three samples, RABV-GEO-19, -95, and -96, only partial genomes were obtained.

Heterozygous SNVs were observed in twenty-four samples, but only three samples were found to have more than three heterozygous SNVs across the genome: RABV-GEO-19, 62, and 95. They were found to have 27, 80, and 77 heterozygous SNVs, respectively. We interpret the relatively large number of heterozygous SNVs in these samples to indicate a potential mixed genotype in each, which may have contributed to the breakage (gaps) in the assemblies for these particular samples. A summary of each observed heterozygous SNV, including depth of coverage at each position and quality scores, can be found in Appendix A.

### 3.3. Phylogeny of the Newly Identified RABV Genomes

To evaluate the diversity among the 77 RABV-GEO genomes successfully assembled, we first examined similarity by a pairwise comparison of the *N* genes, as is consistent with the majority of RABV phylogeny studies. This resulted in a range of 81.82–100% nucleotide identity. Then, to compare the new genomes with other rabies viruses from multiple geographical origins and clades, we created phylogenetic trees for the highly conserved *N* gene [45] as well as the more divergent *G* gene [46]. RABV-GEO-95 was excluded from the *G* tree due to incomplete sequence coverage. In both cases, the alignments that were used for these analyses were generated as described in Materials and Methods, using our data in addition to published sequences collated from the literature ([16,17,21]; Appendix A).

The *N* gene phylogenetic tree shows that the RABV-GEO samples clustered together with reference samples from the Cosmopolitan clade, which represented the available NCBI *N* sequences from Georgia and other proximal countries (Figure 1). As expected from Cosmopolitan samples, they were distant from the Bat, Asian, and Indian Subcontinent clades, while reference samples from the Africa-2, Africa-3, and Arctic-related clades were located midway. Interestingly, while most of the RABV-GEO samples were tightly grouped with reference samples from Georgia and bordering countries, two of the samples, RABV-GEO-37 and -54, branched out with other Cosmopolitan samples from more diverse origins such as Azerbaijan (LN879480), Tajikistan (KY765901), Russia (AY352481), Kazakhstan (AY352489), Hungary (U43025), Bosnia and Herzegovina (U42704; U42706), and Poland (U22840). Samples collected in the Georgian subregions of Samegrelo, Adjara, Shida Kartli, Imereti, Kvemo Kartli, and Kakheti mostly clustered together, with a few exceptions. Samples collected in Tbilisi and Guria did not cluster together clearly.

The *G* phylogenetic tree showed a similar pattern to the one observed in the *N* tree, grouping the RABV-GEO samples with the Cosmopolitan clade where most of the RABV-GEO samples were associated to the reference samples for Georgia and its bordering countries (Figure 2). In addition, RABV-GEO-37 and -54 branched out from the main cluster of RABV-GEO samples and associated with reference samples with more diverse geographical locations such as Azerbaijan (LN879480), Tajikistan (KY765901), Hungary (AF325462), and Poland (AF325464).

### 3.4. Clustering Analysis of RABV Full-Length Genomes

The *N*- and *G*-based phylogenetic trees suggest the close relatedness of the RABV-GEO viruses. Nevertheless, this analysis does not address the possibility of higher dissimilarity levels along the rest of their genomes, for example due to hypermutation or recombination events in loci outside the *N* and *G* genes. To further evaluate the relationship of the 77 RABV-GEO to previously described RABV isolates, we performed a clustering analysis including all of the 2769 full genomes publicly available at NCBI. This analysis was performed using MMSeqs2 [40], a computationally efficient approach to incorporate thousands of full-length genomes and address our question as to whether there were dissimilarities across the genome that may have been missed by examining only *N* and *G* genes from the publicly available RABV sequences. The complete set of RABV sequences were grouped into 124 distinct clusters, and the RABV-GEO samples were assigned to only two of these clusters. Interestingly, all but two samples, RABV-GEO-37 and -54, fell within one cluster. This result recapitulated what we observed in the phylogenetic analysis of *N* and *G* sequences and thus strengthens the notion that all RAVB-GEO viruses are close relatives to each other.

### 3.5. Amino Acid Sequence Variability in the RABV G Ectodomain

The RABV *G* gene encodes a transmembrane glycoprotein containing an ectodomain that protrudes from the viral membrane and is responsible for binding to cellular receptors, mediating viral entry into host cells. RABV G is also the target of both the rabies vaccine and the HRIG component of PEP [47]. Upon analysis of the amino acid composition of the G ectodomain of the RABV-GEO samples, the viral sequences were assigned to 12 groups, ECTO1–ECTO12, based on sequence identity (i.e., members of each ECTO group are composed of viruses with identical ectodomain sequences). The largest group was ECTO1, which included approximately 80% of the total sequenced viruses. Among the other 11 groups, ECTO2 contained about 7% of the sequenced viruses, while the remaining ECTO groups each represented <3% of the total number of samples (Appendix A). Using principal components analysis, we found no significant correlation among ectodomain group assignment and subregion from which samples were collected (Appendix A). Figure 3A shows the sequence of the G ectodomain from viruses of ECTO1, with antigenic sites underlined and sequence variations across the different ECTO groups indicated in red. The specific amino acid changes characteristic of each individual ECTO group are shown in Figure 3B.

### 3.6. Global Sequence Variability of RABV Proteins

In order to have a global overview of the variability of the full complement of the RABV proteins, we performed individual clustering analysis for amino acid sequences of N, P, M, G and, L. We included previously reported sequences obtained from the NCBI database as well as the protein sequences for the viruses described in this work. As can be seen in Table 1, the number of sequences for N far exceeds the reported sequences for the other RABV proteins, followed by G, with P, M, and L significantly behind. This disparity is likely the result of the wider interest of the scientific community in the sequence of N (as a conserved marker used for RABV phylogenetic analysis) and G (used both for phylogenetic analysis and for assessing the likely responses to RABV vaccines and antibody therapy). As expected from its high degree of conservation, N grouped in only two clusters, while P, M, G, and L clustered in nine, four, five, and five clusters, respectively. Furthermore, in agreement with the low stringency used for the protein clustering analysis (see Section 2), RABV-GEO samples grouped in just one cluster for each of the proteins. Representative sequences from the clusters in Table 2 were used for sequence alignment analysis to visualize regions of amino acid conservation all along the protein sequences (Appendix A).

### 3.7. Epitopes in RABV-GEO P, M, and L Proteins

Antibodies that neutralize RABV bind to the surface G glycoprotein, preventing interaction with cellular receptors [47,48,49]. Because the effectiveness of RABV vaccines and therapies depends on efficient neutralization, antibodies directed against G have been more intensely characterized than antibodies against other RABV proteins. However, monoclonal antibodies against P, M, and L have been isolated and characterized, mostly for the purpose of detection and diagnostics [50,51,52]. Additionally, understanding protein homology between distinct RABV isolates is important for evaluating direct-acting antivirals that have been investigated as possible routes for the generation of novel RABV therapeutics. Such approaches include targeting specific matrix protein domains and preventing associations between the phosphoprotein-polymerase complex and N to block the formation of a fully assembled helical ribonucleoprotein complex [53]. To define the conservation of known epitopes located in P, M, and L, we searched in the BV-BRC for rabies epitopes, and then compared them to the protein sequences in the RABV-GEO viruses (Table 2). Of the nineteen epitopes identified in this analysis, two were not present in the selected sequences and six were conserved among the representative sequence for the cluster including RABV-GEO sequences. Each of the epitopes for M (2 epitopes) and L (4 epitopes) were conserved, but 11/13 P epitopes were not conserved.

### 3.8. Neutralization of RABV Isolates by mAb A6

Our previous studies have shown that mAb A6, a human IgG1 raised against the G of Australian bat lyssavirus (ABLV), is broadly neutralizing against a wide array of phylogroup I lyssaviruses [44]. To demonstrate that these phylogroup I Georgian RABV isolates could also be efficiently neutralized by A6, we cultured several distinct viral isolates from the harvested Georgian canine brain tissues. Specifically, we isolated and passaged RABV-GEO-6, -9 and -20, which were among samples with the highest quality tissue and which also yielded robust bands in our initial PCR screening. Viruses from these tissues reached a titer in the range of 10^5^–10^6^ focus-forming units (ffu) after the second passage, and these second-passage viral stocks were thus used for neutralization analysis. As shown in Figure 4, each of these cultured viruses was efficiently neutralized by mAb A6, with a potency similar to our previously reported analyses of other phylogroup I lyssaviruses [44].

## 4. Discussion

In this study, we sequenced and annotated 77 new RABV genomes isolated from rabid dogs and jackals in Georgia. The genomes have been deposited in GenBank (Appendix A). Because the largest numbers of regionally relevant RABV sequences in GenBank are partial genome sequences that include only *N* or *G*, we used these genes as the basis for the phylogenetic analyses presented in this study. The phylogenetic analyses of the *N* and *G* RABV genes indicate that all the samples belong to the Cosmopolitan clade, and they show a strong clustering with other RABV from samples isolated in Georgia and neighboring countries. The similarity among sequences generated through this study is further supported by the cluster analysis of full genomes, which clustered singularly for each of the five proteins analyzed. These results recapitulate the finding that RABV generally clusters by geographic origin [16]. Given metadata on the precise location in the country of Georgia from which each sample was collected, we were further able to investigate the geospatial clustering patterns among our samples, as well as comparing them with the data from a similar study of *N* gene sequences from distinct Georgian samples [21]. Although regionally distinct isolates generally clustered together, two isolates from Tbilisi and Kvemo Kartli (RABV-GEO-37 and -54, respectively) were more diverse, perhaps reflecting a human-facilitated movement of rabid animals from neighboring countries where these genotypes are more common.

The phylogenetic analyses also failed to show a distinction between the RABV strains infecting domestic dogs and those that infect jackals. This result implies that jackals are part of the rabies transmission chain in domestic dog populations, an interpretation that is consistent with the results of other studies [54,55]. Consequently, efforts to eliminate dog rabies in Georgia and other Eastern European countries are unlikely to yield lasting results unless the RABV reservoir in jackals is addressed simultaneously. Furthermore, the continued expansion of golden jackals across much of Eurasia [56] suggests that the control of RABV infection in this species may be a crucial element of rabies elimination in the entire region. Indeed, the authors of a phylogenetic study of Middle Eastern RABV strains in dogs and wildlife came to a similar conclusion [57].

Amino acid sequence analyses of the G protein ectodomain revealed a high degree of conservation in the antigenic sites. Indeed, among the 12 ECTO groups, we identified only one amino acid change within the RABV G antigenic sites: R264H in the ECTO2 group, involving the C-terminal residue within the antigenic site G5. This change has been previously reported in strains of an Asian origin [58,59]. Notably, this amino acid is not critical for recognition by mAb AR16, and both His and Arg residues are common at position 264 in RABV *G* genes from a variety of sequenced isolates [60]. However, although R264H does not impair the binding of mAb AR16, it does impede the binding of other anti-RABV G mAbs, such as 1D1 and 6-15C4 [61,62], demonstrating that this amino acid substitution does impact the spectrum of antibodies capable of interacting with this site. In consideration of these facts, and because mAb cocktails are now being approved for PEP after human RABV exposures [63], it would be optimal to choose mAbs for PEP formulations that are relatively insensitive to the R264H substitution, in order to ensure efficacy against strains with this amino acid variation.

The remainder of the residue changes among the ECTO groups do not alter putative glycosylation sites (N-X-S/T) or disulfide bonds and will require further analysis to evaluate putative effects on the tertiary structure of the G protein. In this regard, the I90T and S156G amino acid changes found in the ECTO6 group may be of particular interest, since this group contains RABV-GEO-37 and -54, the more phylogenetically distant samples identified in this study. It would be interesting to determine if these and other residue changes outside of the antigenic sites have significant effects on parameters such as viral titer or the efficiency of cellular infection. Overall, a sequence analysis of *G* genes from RABV-GEO isolates suggests that current vaccines should offer protection against recently circulating Georgian canine strains.

As expected, human mAb A6 neutralized three RABV-GEO isolates that we were able to culture successfully. These data support the conclusion that mAb A6 is highly efficient in the neutralization of lyssaviruses across phylogroup I [44], including currently circulating strains. Given the in vitro potency of this mAb, the utility of A6 for in vivo applications (e.g., as part of a PEP cocktail [63] or as a therapeutic [64]) should be explored.

Overall, our results show that Georgian RABV strains exhibit a high degree of phylogenetic relatedness, with most strains contained within a single cluster. As mutations in the RABV glycoprotein antigenic sites are uncommon, it is highly likely that RABV vaccine strains provide strong protection against the Georgian RABV strains currently circulating in dogs and jackals. Because we observed no distinction in RABV strains between dogs and jackals, the elimination of dog rabies is unlikely to be stably achieved through the vaccination of dogs alone. Indeed, the successful elimination of rabies in Western European countries ultimately required a major international effort to control fox rabies through oral vaccination [65]. A similar program focused on jackals in Eastern Europe (and Western Asia) will likely be required as an adjunct to the vaccination of domestic dogs to eliminate rabies in this region.

## Figures and Tables

**Figure 1 viruses-15-01797-f001:**
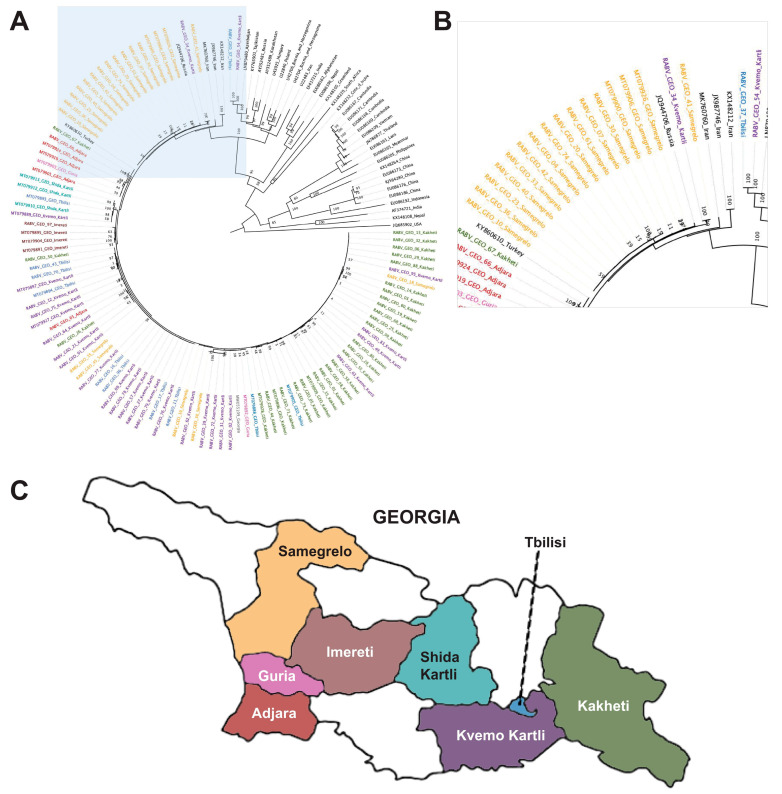
Phylogenetic tree of Georgian lyssavirus *N* gene sequences. A total of 140 sequences with 1353 nucleotide sites were included, and the log-likelihood of the consensus tree was −9845. The tree is rooted at the midpoint. The labels of samples collected in Georgia with geographical metadata are colored by subregion: Kakheti (green), Kvemo Kartli (purple), Samgrelo (orange), Tbilisi (blue), Guria (pink), Adjara (red), Shida Kartli (teal). (**A**) Full phylogenetic tree (the area shadowed in blue is zoomed in the B panel). (**B**) Zoomed in view of the branching area of the tree containing the RABV-GEO samples. (**C**) Map of the country of Georgia, indicating regions from which samples were collected (adaptation of “File: Regions of Georgia (country).svg” by Nordwestern, licensed under CC BY-SA 4.0). Colors were coded to match samples in (**A**,**B**).

**Figure 2 viruses-15-01797-f002:**
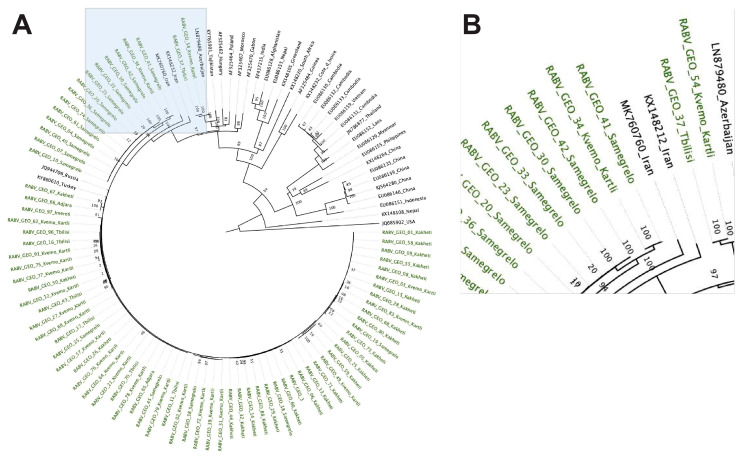
Phylogenetic tree of Georgian lyssavirus *G* gene sequences. A total of 110 sequences with 1578 nucleotide sites were included, representing the available NCBI *G* sequences relevant to geographic/clade distribution. The log-likelihood of the consensus tree was −12,159. The tree is rooted at the midpoint. The labels of samples collected in this study are colored green. (**A**) Full phylogenetic tree (the area shadowed in blue is zoomed in the **B** panel). (**B**) Zoomed in view of the branching area of the tree containing the RABV-GEO samples.

**Figure 3 viruses-15-01797-f003:**
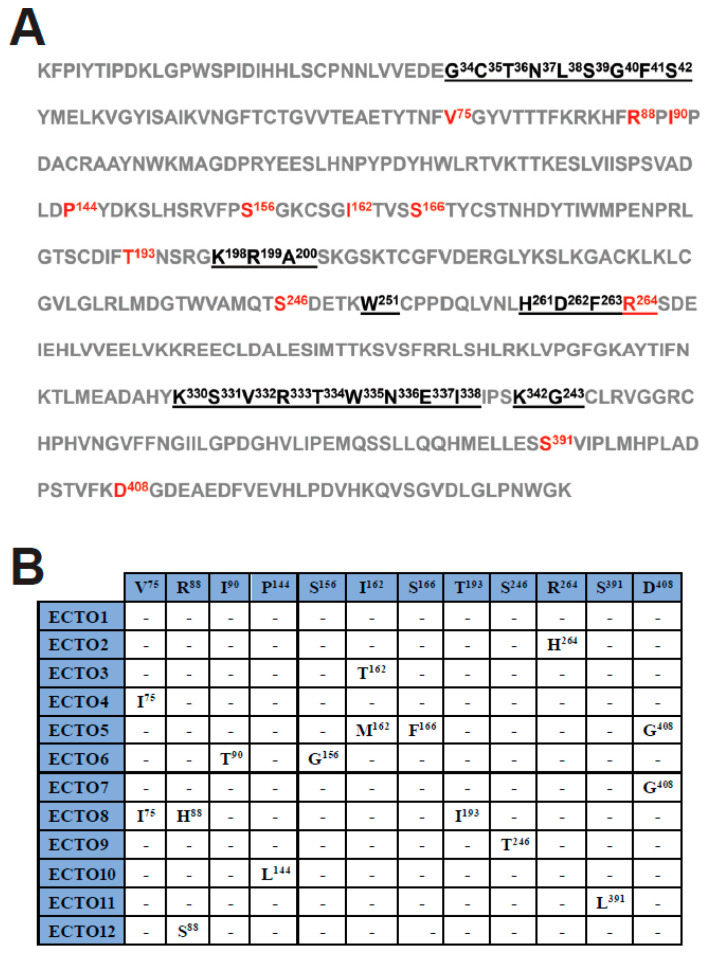
Amino acid sequence of the G protein ectodomain of the Georgian ECTO1 group. (**A**) The amino acid sequence of the RABV G ectodomain is shown, using the sequence derived from the viruses of the ECTO1 group. The antigenic sites are underlined. Amino acids that vary between different ECTO groups are indicated in red. (**B**) Amino acid changes in different ECTO groups with respect to the reference (ECTO1). The positions of the amino acids in the ectodomain are indicated in superscript.

**Figure 4 viruses-15-01797-f004:**
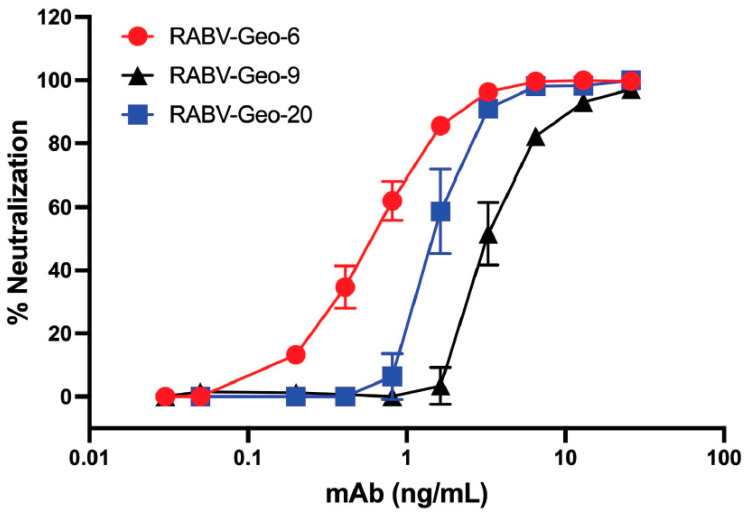
Neutralizing activity of anti-ABLV G human mAb A6 against RABV-GEO isolates. RABV-GEO viruses 6, 9, and 20 were isolated from primary canine brain tissue, followed by one round of in vitro passaging. Virus neutralization was measured by incubation of viral supernatants with human mAb A6, followed by infection of N2a cells. Viral foci were identified by staining with anti-Rabies G. Graph shows percent virus neutralization over a series of 5-fold dilutions of mAb A6.

**Table 1 viruses-15-01797-t001:** Cluster analysis of RABV-GEO N, P, M, G, and L proteins.

Protein	Number of Sequences from NCBI	RABV-GEO Sequences	Number of Clusters	Number of Clusters Containing RABV-GEO Samples
N	7489	77	2	1
P	2579	77	9	1
M	2178	77	4	1
G	5915	77	5	1
L	2948	77	5	1

**Table 2 viruses-15-01797-t002:** Linear peptide epitopes within L, M, P among representative amino acid sequences. Differences are shown in bold red.

Epitope ID	Epitope Sequence	Difference in Representative Sequence (If Relevant, Colored Red and Bold)	Protein Name	Protein ID	Protein Accession	Start	End
1336019	EIFSIP	-	L	ADJ29912.1	P11213	1479	1484
1336142	RALSK	-	L	ADJ29912.1	P11213	1659	1663
1336215	VFNSL	-	L	ADJ29912.1	P11213	1724	1728
93714	RKLGWWLKL	not present in representative sequence set	L	SRC266014	P11213		
929568	PPDDD	-	M	CEH11416.1	P08671	42	46
929569	PPYDDD	not present in representative sequence set	M	ADJ29910.1	P08671	25	30
12638	EKDDLSVEAEIAHQIA	E**E**DDLSVEAEIAHQIA	P	P69479.1	P06747	191	206
1795	AHLQGEPIEVDNLPEDMKRLQLDDKKPSGL	AHLQGEPIEVDNLPEDM**R**RL**N**LDD**G**K**SPN**L	P	AAK54996.1	P06747	37	66
20735	GKYREDFQMDEGDPS	GKYREDFQMDEG**EDP**	P	SRC279966	P06747		
23111	GVQIVRQIRSGERFLKIWSQ	GVQIVRQ**M**RSGERFLKIWSQ	P	NP_056794.1	P06747	101	120
31389	KIPLRCVLGWVALANSKKFQLLVEADKLSKIMQDDLNRYTSC	K**L**PLRCVLGWVALANSKKFQLLVEADKLSRIMQDDLNRY**A**S**S**	P	AAZ07892.1	P06747	256	297
31531	KKETTSISSQRDSQSSKA	KKETTS**TP**SQR**E**SQSSKA	P	AAK54996.1	P06747	154	171
38005	LMDEGEDPSLLFQSYLDNVGVQIVRQMRSGER	**Q**MDEGEDPSLLFQSYLDNVGVQIVRQMRSGER	P	AAK54996.1	P06747	82	113
451183	VLGWV	-	P	AAK55085.1	P06747	262	266
53736	RFLKIWSQTVEEIISYVAVN	RFLKIWSQTVEEIISYV**T**VN	P	P69479.1	P06747	113	132
59254	SLLFQSYLDNVGVQIVRQIR	SLLFQSYLDNVGVQIVRQ**M**R	P	NP_056794.1	P06747	90	109
68113	VEAEIAHQI	-	P	P15198.1	P06747	197	205
93760	RQMKSGGRF	RQM**R**SG**E**RF	P	AAY23584.1	P06747	68	76
94299	YLDNVGVHI	YLDNVGV**Q**I	P	AAK55014.1	P06747	96	104

## Data Availability

Complete viral genome sequences are available in GenBank, with IDs OQ603609 to OQ603685.

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
