# Peer review of "Genomic Surveillance of Rabies Virus in Georgian Canines"

_viruses, 2023, doi:10.3390/v15091797_

Round 1
Reviewer 1 Report
Overall this is a paper that, if improved, provides new information on rabies in Georgia, addressing a gap in research. The introduction needs some background information on the context and a better explanation on why this research is worthy of publication, i.e. what gap is it filling.
The methods are very long and need more consistent structure and justification at places.
The results are also very long and many parts are not discussed in the Discussion. Please consider a better flow and potentially moving some results to supplementary material for an easier read.
Discussion is very short, does not touch on many results, does not discuss the context around the results adequately and has not conclusion.
Please find specific comments in the pdf attached.

Quality of English is adequate.
Author Response
We very much appreciate the detailed and thoughtful critiques from the reviewers regarding our manuscript. We have endeavored to address reviewer critiques as completely and thoroughly as possible. Our specific responses are in bold type, after each reviewer statement (indicated by italics):
REVIEWER 1
Overall this is a paper that, if improved, provides new information on rabies in Georgia, addressing a gap in research. The introduction needs some background information on the context and a better explanation on why this research is worthy of publication, i.e. what gap is it filling.
The methods are very long and need more consistent structure and justification at places.
The results are also very long and many parts are not discussed in the Discussion. Please consider a better flow and potentially moving some results to supplementary material for an easier read.
Discussion is very short, does not touch on many results, does not discuss the context around the results adequately and has not conclusion.
Please find specific comments in the pdf attached.
Critiques are addressed below, in the specific comments taken from the marked-up PDF.
REVIEWER 1 COMMENTS FROM MARKED-UP PDF
Abstract:
- Word change eradicating vs. eliminating:
The requested change has been made
- Clarify human or dog vaccines
This has been clarified as human
Introduction:
- Comment “highly successful domestic dog vaccination programs” are not just for dogs
Text changed from “dog” to “animal”
- useful to elaborate on potential reasons why somewhere in the paper sporadic cases of dog-mediated rabies still occur in Eastern Europe
We agree, and we have now added text to address this important point both in the Introduction and in the Discussion
- Strand à stranded RNA
The requested correction has been made
- “Although a few studies have provided complete sequences of a large number of isolates” – why is it important to have more isolates and more complete genomes? Add some thoughts in intro to explain significance of work
- Give some info on what is available in Eastern Europe and specifically Georgia to really set the scene and explain that there is a gap you are trying to fill with this paper
To address the above two comments, we have added text to paragraphs 2 and 5 of the introduction
Materials and methods:
- When describing your methods add some rational for the use of each, especially the partial sequencing followed by the complete genome sequencing. it will make it easier for readers who are not experts in the field
We have added the requested rationale to the beginning of paragraph 2 in Methods, 2.2 and the beginning of paragraph 1 in Methods, 2.4
- Say who runs the rabies surveillance program in Georgia
This clarification has been added to section 2.2
- How were N gene sequences selected for the Geneious Tree Build?
These were the available NCBI N sequences from Georgia and other proximal countries. This information has been added to section 2.3.
- How was the global alignment performed?
We have edited the text in section 2.3 to be more accurate in stating how the tree-building was performed, without reference to a global alignment.
- Comment on Sequencing data processing and genome assembly: why do you have separate bioinformatics/analysis here, yet before you had it all together in a single section?
In the interest of clarity and reproducibility for the computational work performed with our high throughput sequencing data, the authors would prefer that these sections to remain separated into distinct sections that outline each key analysis step. By contrast, the much simpler initial Sanger sequencing of PCR products can be described more succinctly. Thus, it is appropriate to leave this description in a single methods section.
Results:
Section 3.3 phylogeny:
- Why were only N genes compared using pairwise comparison?
In many genomic studies of RABV, N is the only gene compared. This comparison is consistent with the RABV literature and helps to place our data among the existing sequences, many of which are only for the nucleoprotein gene. We have updated the first sentence of this section to mention that this is a consistent with the literature.
- Why did you do complete sequence and then analyze them separately? it's important to explain your rational
Our interpretation of this comment is that Reviewer 1 is asking why separate trees were generated for both N and G rather than simply using the complete genomes, since we obtained the complete genomes for nearly all samples. The rationale for this is 1) the greatest numbers of regionally relevant sequences from RABV were for N, which is highly conserved, and 2) the more divergent G gene was analyzed using available sequences to provide orthogonal confirmation of clade and evaluate the relationship of the samples from our study to other RABV G sequences in the public database. An abbreviated version of this rationale has been added to the first paragraph of the discussion.
- Clarify what we used sequences from the literature for…
Our interpretation is that Reviewer 1 is asking for clarification about our alignments for both N and G trees. These alignments were generated using our data in addition to the reference sequences that were collated from the RABV literature. Those sequences were published in NCBI, with accession numbers listed in Supplementary Table 3 and the references cited in text. We have updated the text to state that “In both cases, alignments that were used for these analyses were generated as described in the methods using our data in addition to published sequences collated from the literature ([11,12,41]; Supplementary Table 3).
- Add comment in the discussion about how GEO-37 and GEO-54 branched out with other Cosmopolitan samples from more diverse origins
The authors have addressed this point in the first paragraph of the discussion: “Although regionally distinct isolates generally clustered together, two isolates from Tbilisi and Kvemo Kartli (RABV-GEO-37 and -54, respectively) were more diverse, perhaps reflecting human-facilitated movement of rabid animals from neighboring countries where these genotypes are more common.” The authors believe any statements beyond the above would be overly speculative.
- Last comment in 3.3 text asks the geographic metadata to be added to the tree
The reviewer has requested geographic metadata be added to the tree. This information is available in Figure 1 text description of the color-coding. Metadata including country, sample collection year, host species, clade, GenBank ID, and subregion (if Georgia) are available in Supplemental Table 3. The nature of the host species and lack of correlation based on our analysis (Supplementary Figure 2) is discussed in the text.
Figure 1:
- How were the 140 N sequences selected
These are relevant N sequences available from NCBI for the region that have been described in the literature. This rationale has been added to the text of section 3.3, paragraph 2.
- B) why zoom in?
The figure is zoomed in to provide clarity for the region where the tree branches into the relevant RABV-GEO cluster. Some correlation by region is visible due to this major branch out from RABV sequences from other countries. The two differing samples, RABV-GEO-37 and -54, are more clearly visible in a separate branch in the zoomed-in view.
- C) indicate source of map data
The Creative Commons source is now provided
Figure 2:
- How were 110 G sequences chosen
These sequences were selected based on availably from the literature and selected to cover relevant geographic/clade distribution. This rationale has been added to the text of the Fig 2 legend.
- Why is the color coding different between Figures 1 and 2
Figure 1 color coding is in accordance with subregion of Georgia to identify how the available data cluster with publicly available sequences collected from the same subregion of Georgia; however, G sequences from Georgia were not available in the public database and therefore we did not color code in the same way.
Section 3.4
- How were the 2,769 full genomes selected for clustering analysis?
These are all the RABV genomes that are available in NCBI when filtering for full-length genomic RNA, as described in methods section 2.9.
- Request for a supplemental tree with metadata “to understand where the rest of the sequences are coming from.”
While we decided to perform clustering analysis using full-length genomes after examining our phylogenetic trees based on N and G, this analysis differs from phylogenetic analysis and there is no resultant tree to represent these data. Both analyses bear similarities in that phylogenetic analysis and clustering analysis produce “clusters” by which to sort and understand the data among groups. MMseqs2 stands for “Many-against-Many sequence searching” and is designed to search and cluster large protein and nucleotide sequence sets. While it is indeed possible to align thousands of complete genomes to perform phylogenetic analysis, it is also computationally expensive, and the resultant trees may not be informative without zooming in extensively due to the large number of branches/clusters. MMSeqs2 is a computationally efficient and comparable alternative to phylogenetic analysis by which we answered our question “is there other dissimilarity across the genome that we’ve missed by looking only at where these samples sit in relation to other N and G genes from publicly available RABV sequences?” Section 3.4 has been revised to provide better clarity about the work that was performed.
- What defines a cluster?
As described in the methods section, MMSeqs2 “target clustering mode” was utilized so that partial sequences were heavily weighted against being representative of a cluster and a minimum of 0.95 nucleotide identity was required for full length genomes to fall within the same cluster.
- Where were the rest of the full genomes samples from?
The authors believe that Reviewer 1 is asking for the source of the full genomes that were used in MMSeqs2 clustering. As described in the methods section 2.9: full length genomes of RABV were downloaded from NCBI, taxID 11292, Lyssavirus rabies, accessed March 2023.
- If you run a tree with just your full genomes would you get more clusters?
The authors believe that Reviewer 1 is asking what would happen if only the samples from this study were evaluated using MMSeqs2, in the absence of full-length genomes from NCBI. This exercise results in 12 clusters with the majority of samples falling within a single cluster. This result recapitulates the finding that the genomes characterized through this study are similar but not identical, as already demonstrated by the short sub-branches zoomed in for both Figures 1 and 2, and thus is not included in the manuscript because it does not add any new information.
Discussion:
- 1st paragraph: It's not clear that you have compared whole genomes with others from Georgia or the region
The text has been clarified to say “The similarity among sequences generated through this study is further supported by cluster analysis of full genomes, which clustered singularly for each of the five proteins analyzed.” Again, the majority of RABV sequences available from Georgia are for the N gene only; thus, comparisons at the whole genome level for Georgia are not possible. We believe this fact is now made clear by other changes to the text, outlined above.
- 2nd paragraph: practical implication of R264H impeding binding of other anti-RABV G mAbs 1D1 and 6-15C4?
Based on the scientific literature, the R264H change in the ECTO2 group of viruses is predicted to impede the binding of mAbs 1D1 and 6-15C4, although to demonstrate this, tested empirical testing is required. Note that we have also corrected the references in this relevant paragraph, which were misnumbered. We have added a statement of the practical implications to this paragraph
- Comment at the end: you have presented a massive amount of results, yet your discussion is very short, does not include a concluding paragraph and does not explain to the reader the wider context or why this study is important to be published.
We have added more discussion to better emphasize the importance of this work, as well as a concluding paragraph.
Reviewer 2 Report
The manuscript "Genomic Surveillance of Rabies Virus in Georgian Canines" reported the study of the circulation trend of Rabies viruses with the genomic sequence of 77 samples from Georgian canines and the phylogeny with other sequences from NCBI. The study analyzed the genome sequences with appropriate methods/tools, and the results supported the goal of indicating the recent circulation rend on Georgian canines. It is a valuable addition to understanding the circulation trend and phylogeny. It also serves the goal of surveillance of rabies to assure the protection efficacy of current rabies vaccines.
There are a few statements that are not very concise and need to change. A few of the methods and result presentation also need to improve. I'm providing some suggestions here. Unfortunately, there are no line numbers in the manuscript I received, so I have to quote the sentences so expect the review will be longer.
Introduction part
1.) "These deaths are attributable to a combination of poor access to PEP, the absence of effective treatments for symptomatic rabies,...." so far, there are less than 20 cases of human survival documented. There are really no effective treatments for rabies, but PEP and dog vaccination are two effective preventions, so this sentence should change to two sentences.
2.) "Lyssaviruses are negative-strand RNA viruses of the family Rhabdoviridae..." "Lyssavirus genomes encode five genes, N (nucleoprotein), P (phosphoprotein), ..."
While the rabies virus belongs to the genus Lyssavirus, the lyssavirus genus includes a few other viruses, so it is not exchangeable between the lyssavirus and rabies virus. As the manuscript was talking about rabies, I suggest just mentioning rabies belonging to lyssavirus and then describing all characteristics of rabies viruses instead of lyssavirus, as it can be very confusing.
Method part
1) Rabies virus is a dangerous virus. Inhalation of aerosolized rabies virus can be a potential non-bite route of exposure to researchers who process the samples. I would suggest authors mention the biosafety cautions when sampling and process specimens as a reminder to the research community of rabies.
2) Best-fit model for phylogenetic analysis can be tested with different methods. Please confirm if you are using IQ-TREE's model finder or other methods. IQ-TREE should not be spelled as iqtree. Please correct it.
3) "2.5. Analysis of heterozygous single nucleotide variants (SNVs)" mentioned, "with a minimum of 80% identity and evaluated for heterozygous SNVs using CLC Genomics Workbench v23 (QIAGEN, Hilden, Germany) with a minimum frequency of 35% and a minimum coverage of 10". There are no citations in this part. Can you please explain how you set up these criteria for the analysis and why?
Result part:
1) what is Table 1 for? There is no description of the content of Table 1. I was confused by listing the number of sequences from NCBI. Did you mean you analyzed your 77 sequences with thousand of NCBI sequences? There are no results (trees) of this analysis in the supplementary materials.
2)3.3. Phylogeny of the newly identified RABV genomes
"In both cases, we used reference sequences from selected samples previously described" Can you clarify the standard you used to choose the reference sequences? In addition, I suggested replacing Table 1 with supplementary Table 3 as supplementary Table 3 has important information for readers to understand what reference sequences were used in the analysis.
3) "3.8. Neutralization of RABV isolates by mAb A6."
"To demonstrate that these phylogroup I Georgian RABV isolates could also be efficiently neutralized by A6, we cultured several distinct viral isolates from the harvested Georgian canine brain tissues. Specifically, we isolated and passaged RABV-GEO-6, -9 and -20." Can you explain what criteria you used to choose these three samples to culture? In the cluster part, you mentioned that two samples, GEO-37 and 54, are in one cluster, and all the other samples are in a different cluster. I suggest at least including representative isolates from both clusters in your Neutralization assay to better support your conclusion that mAb A6 is highly efficient in the neutralization of currently circulating strains.
Discussion Part
1) "As expected, human mAb A6 neutralized two RABV-GEO isolates that we were able to culture successfully." As I see, you did a neutralization assay for three isolates with similar results. Why did you state two isolates in the discussion?
Please see the suggestions for authors
Author Response
We very much appreciate the detailed and thoughtful critiques from the reviewers regarding our manuscript. We have endeavored to address reviewer critiques as completely and thoroughly as possible. Our specific responses are in bold type, after each reviewer statement (indicated by italics):
REVIEWER 2
Comments and Suggestions for Authors
The manuscript "Genomic Surveillance of Rabies Virus in Georgian Canines" reported the study of the circulation trend of Rabies viruses with the genomic sequence of 77 samples from Georgian canines and the phylogeny with other sequences from NCBI. The study analyzed the genome sequences with appropriate methods/tools, and the results supported the goal of indicating the recent circulation rend on Georgian canines. It is a valuable addition to understanding the circulation trend and phylogeny. It also serves the goal of surveillance of rabies to assure the protection efficacy of current rabies vaccines.
There are a few statements that are not very concise and need to change. A few of the methods and result presentation also need to improve. I'm providing some suggestions here. Unfortunately, there are no line numbers in the manuscript I received, so I have to quote the sentences so expect the review will be longer.
Introduction part
1.) "These deaths are attributable to a combination of poor access to PEP, the absence of effective treatments for symptomatic rabies,...." so far, there are less than 20 cases of human survival documented. There are really no effective treatments for rabies, but PEP and dog vaccination are two effective preventions, so this sentence should change to two sentences.
The requested changes have been made.
2.) "Lyssaviruses are negative-strand RNA viruses of the family Rhabdoviridae..." "Lyssavirus genomes encode five genes, N (nucleoprotein), P (phosphoprotein), ..."
While the rabies virus belongs to the genus Lyssavirus, the lyssavirus genus includes a few other viruses, so it is not exchangeable between the lyssavirus and rabies virus. As the manuscript was talking about rabies, I suggest just mentioning rabies belonging to lyssavirus and then describing all characteristics of rabies viruses instead of lyssavirus, as it can be very confusing.
The requested changes have been made.
Method part
1) Rabies virus is a dangerous virus. Inhalation of aerosolized rabies virus can be a potential non-bite route of exposure to researchers who process the samples. I would suggest authors mention the biosafety cautions when sampling and process specimens as a reminder to the research community of rabies.
We have added to Methods, 2.1 Biosafety Precautions
2) Best-fit model for phylogenetic analysis can be tested with different methods. Please confirm if you are using IQ-TREE's model finder or other methods. IQ-TREE should not be spelled as iqtree. Please correct it.
The best-fit model was used with IQ-TREE and the appropriate references for this are included. The text has been corrected from iqtree to IQ-TREE.
3) "2.5. Analysis of heterozygous single nucleotide variants (SNVs)" mentioned, "with a minimum of 80% identity and evaluated for heterozygous SNVs using CLC Genomics Workbench v23 (QIAGEN, Hilden, Germany) with a minimum frequency of 35% and a minimum coverage of 10". There are no citations in this part. Can you please explain how you set up these criteria for the analysis and why?
This manual analysis was performed using CLC Genomics Workbench v23, license purchased from QIAGEN as described in the methods section, and not a program referrable from the literature.
As described in Methods section 2.4, not all samples were assembled to a single contig via de novo assembly and several required manual genome closure. While this is generally acceptable, we were curious as to the reason. We continue to explain in Methods section 2.5 that the purpose of analyzing heterozygous SNVs was to evaluate the possibility of coinfection with multiple genotypes or lineages of rabies virus. In other words, because we did not obtain complete genomes for select samples we wanted to know if it was perhaps due to the existence of low-level variants, or quasispecies, within those particular samples that could have interrupted and therefore resulted in breakage of the assemblies. The parameters selected are default settings in CLC Genomics Workbench v23 and loose enough to identify multiple genotypes/lineages of RABV without permitting irrelevant sequences to map to the reference. The minimum frequency of 35% and minimum coverage of 10 ensures that we could not draw conclusions from regions of the genome with poor coverage or from SNVs that could be due to sequencing error, for example.
Result part:
- what is Table 1 for? There is no description of the content of Table 1. I was confused by listing the number of sequences from NCBI. Did you mean you analyzed your 77 sequences with thousand of NCBI sequences? There are no results (trees) of this analysis in the supplementary materials.
Thank you for bringing this formatting issue to our attention. Table 1 is referred to in section 3.6. We request that the editors ensure that Table 1 is moved to section 3.6, where the callout appears. Indeed, amino acid sequences of N, P, M, G, and L from our 77 sequences were analyzed along with thousands of NCBI sequences; this work is described in the text and section 3.6 as well as in methods section 2.9. The table describes the results as this analysis was a clustering analysis performed using MMSeqs2 and not an alignment of thousands of sequences subsequently used to generate phylogenies (trees).
2)3.3. Phylogeny of the newly identified RABV genomes
"In both cases, we used reference sequences from selected samples previously described" Can you clarify the standard you used to choose the reference sequences?
We have referenced the literature from which these reference sequences were collated. The sequences were collated to represent various clades and a relevant geographic distribution for our work.
In addition, I suggested replacing Table 1 with supplementary Table 3 as supplementary Table 3 has important information for readers to understand what reference sequences were used in the analysis.
In the interest of keeping results concise, we prefer to keep the long Supplementary Table 3 as supplemental material.
3) "3.8. Neutralization of RABV isolates by mAb A6."
"To demonstrate that these phylogroup I Georgian RABV isolates could also be efficiently neutralized by A6, we cultured several distinct viral isolates from the harvested Georgian canine brain tissues. Specifically, we isolated and passaged RABV-GEO-6, -9 and -20." Can you explain what criteria you used to choose these three samples to culture?
We have added text to clarify the selection criteria.
In the cluster part, you mentioned that two samples, GEO-37 and 54, are in one cluster, and all the other samples are in a different cluster. I suggest at least including representative isolates from both clusters in your Neutralization assay to better support your conclusion that mAb A6 is highly efficient in the neutralization of currently circulating strains.
While we agree that such experiments would be beneficial, only a minority of tissue samples in this set were actually of adequate quality to recover virus efficiently. GEO-37 and -54 were unfortunately not among this class of high quality tissues.
Discussion Part
1) "As expected, human mAb A6 neutralized two RABV-GEO isolates that we were able to culture successfully." As I see, you did a neutralization assay for three isolates with similar results. Why did you state two isolates in the discussion?
Our apologies: the neutralization data for the third isolate were collected after the initial draft had been prepared. We simply forgot to update the relevant text in the discussion. This has now been corrected.